# CONTRASTIVE DECODING IMPROVES REASONING IN LARGE LANGUAGE MODELS

## ABSTRACT

We demonstrate that Contrastive Decoding – a simple, computationally light, and training-free text generation method proposed by Li et al 2022 – achieves large out-of-the-box improvements over greedy decoding on a variety of reasoning tasks. Originally shown to improve the perceived quality of long-form text generation, Contrastive Decoding searches for strings that maximize a weighted difference in likelihood between strong and weak models. We show that Contrastive Decoding leads LLaMA-65B to outperform LLaMA 2, GPT-3.5 and PaLM 2-L on the HellaSwag commonsense reasoning benchmark, and to outperform LLaMA 2, GPT-3.5 and PaLM-540B on the GSM8K math word reasoning benchmark, in addition to improvements on a collection of other tasks. Analysis suggests that Contrastive Decoding improves over existing methods by preventing some abstract reasoning errors, as well as by avoiding simpler modes such as copying sections of the input during chain-of-thought. Overall, Contrastive Decoding outperforms nucleus sampling for long-form generation and greedy decoding for reasoning tasks, making it a powerful general purpose method for generating text from language models.

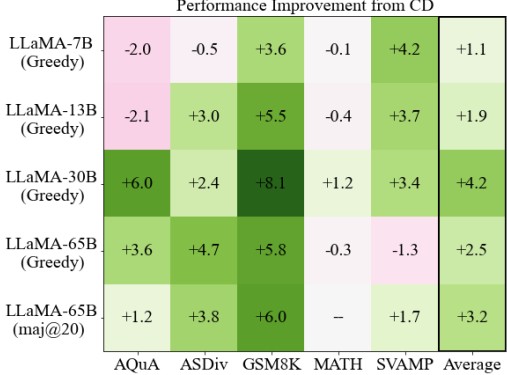

Figure 1: Contrastive decoding improves reasoning across model scales and reasoning tasks.

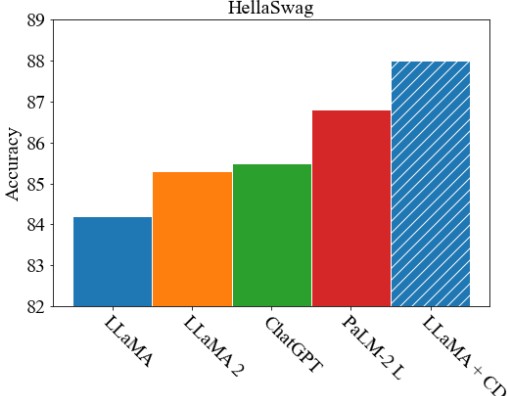

Figure 2: Contrastive scoring significantly improves performance on HellaSwag, a standard commonsense reasoning benchmark.

## 1 INTRODUCTION

Text is generated from large language models (LLMs) in different ways for different tasks. For open-ended text generation tasks, truncated sampling is normally used, as the most likely strings under a model tend to be short and uninteresting (Holtzman et al., 2020). For reasoning problems, greedy decoding is normally preferred, to avoid risking sampling errors. This bifurcation is undesirable; for example it increases the likelihood of reasoning errors during open-ended generation.

We explore the use of Contrastive Decoding (Li et al., 2022) for solving reasoning problems with LLMs. Contrastive Decoding (CD) searches for strings that maximize a weighted difference in

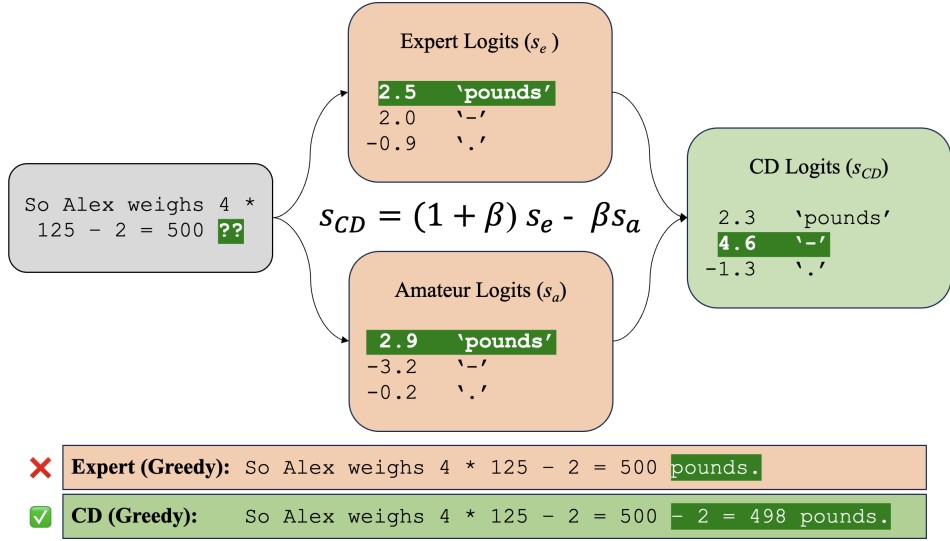

Figure 3: CD accentuates what the expert model has learned that the amateur model has not. Results are taken from greedy decoding with a 65B parameter expert, using $\alpha = 0.1$, $\beta = 0.5$ for CD.

likelihood between a stronger *expert* and a weaker *amateur* model, and was shown to outperform existing methods for open-ended text generation. It achieves this by avoiding undesirable modes of the expert model's distribution, such as short or generic strings, which tend to be the most likely under any model, including the amateur.

We show that Contrastive Decoding outperforms greedy decoding on reasoning problems. On GSM8K, a widely used benchmark consisting of grade-school word math problems, contrastive decoding improves the performance of various LLaMA models by up to 8 absolute percentage points. This result outperforms LLaMA 2, which has 5 billion more parameters and is trained on 40% more data. On HellaSwag, using the CD objective to rank answers leads LLaMA to outperform all existing models except GPT-4. We find general improvement on arithmetic reasoning and multiple-choice ranking tasks, including on models as large as LLaMA-65B, suggesting that Contrastive Decoding could bring such widespread improvements to much larger models.

We also analyze the cause of the improvement from Constrastive Decoding. Empirically, we find that Contrastive Decoding performs less surface-level copying from the prompt than greedy decoding and misses fewer reasoning steps. This result suggests that, similarly to findings in Li et al. (2022), Contrastive Decoding works by reducing repetitive or other undesirable modes of the model distribution. Our current method yields mixed results for commonsense reasoning tasks and slightly degrades factual retrieval, both trends that encourage further refinement of the method.

Overall, we show that Contrastive Decoding not only substantially improves LLM accuracies on a range of benchmarks, but is also the first generation algorithm to achieve state-of-the-art results in both reasoning and text generation problems. These results allow a more unified method for improving generation from language models across tasks.

## 2 CONTRASTIVE DECODING

### 2.1 SIMPLIFIED FORMULATION

The original Contrastive Decoding formulation from Li et al. (2022) explicitly chooses two parameters: $\alpha$ and the intermediate temperature of the amateur distribution $\tau_a$, with the intermediate temperature of the expert fixed at $\tau_e = 1$. We slightly refactor the hyperparameter choice to be more interpretable and simplify the algorithm by working directly in logit space.

Let $s_a^{(i)}$ and $s_e^{(i)}$ be the unnormalized scores (logits) assigned to token $i$ by the amateur and expert models, respectively. $\alpha$ is the same hyperparameter in the original paper: a proportion of the

maximum probability assigned by the expert model, with any tokens assigned a lower probability masked out. $\beta$ is a hyperparameter corresponding to the strength of the amateur penalty. We include a leading $(1 + \beta)$ coefficient to the expert logits to decouple the strength of the contrastive penalty from the expected scale of the output logits, cleanly delineating between the contrastive tradeoff and the final sampling temperature. This matches the formulation of DExperts (Liu et al., 2021), with the expert model serving both as the base prior and steering expert.

$$
\begin{array}{c}
\textbf{1. Determine } \alpha\text{-mask.} \\[4pt]
V_{valid} = \{j \in V, s_e^{(j)} \geq \log \alpha + \max_{k \in V} s_e^{(k)}\} \\[2pt]
\hline \\[-6pt]
\textbf{2. Subtract amateur logits.} \\[6pt]
s_{CD}^{(i)} = \begin{cases} (1 + \beta)s_e^{(i)} - \beta s_a^{(i)} & i \in V_{valid} \\ -\infty & i \notin V_{valid} \end{cases}
\end{array}
$$

A PyTorch implementation for this formulation, as well as the original, can be found in subsection A.1 of the appendix. Our implementation takes three lines of readable code.

## 2.2 Probabilistic Interpretation

Our implementation of $\alpha$-masking has the same interpretation as in Li et al. (2022), given that the expert temperature is fixed to $\tau_e = 1$. We show the equivalence in Appendix A.2.

Further, we can consider the post-softmax probabilities produced by CD as a perturbation of the probabilities predicted by the expert model. Not including $\alpha$-masking, the probability assigned to token $i$ by CD is a normalized adjustment of the probability assigned by the expert model:

$$
p_{CD}^{(i)} \propto p_e^{(i)} \left( \frac{p_e^{(i)}}{p_a^{(i)}} \right)^{\beta} \tag{1}
$$

It is therefore clear that as $\beta \to 0$ the contrastive penalty disappears, and as $\beta \to \infty$ the distribution collapses to the argmax of $p_e^{(i)}/p_a^{(i)}$, which is the original formulation from Li et al. (2022).

## 3 Experiments

### 3.1 Experimental Setup

**Models.** We use untuned models from the LLaMA 1 family (Touvron et al., 2023) at all scales. Unless otherwise stated, we use an untuned LLaMA-65B as the expert and an untuned, LLaMA-architecture model with 1.5B parameters trained on the same data as the other LLaMA 1 models as an amateur. For one ablation study, we use models from the FLAN-T5 family (Chung et al., 2022).

**Decoding Parameters.** We set $\beta = 0.5$ and $\alpha = 0.1$ for all experiments unless otherwise stated. We use greedy decoding, except for self-consistency experiments for which we sample at $\tau = 0.7$ following Touvron et al. (2023).

**Prompting.** For generation tasks, we use 8-shot chain-of-thought prompting, in line with Touvron et al. (2023). The examples are the same as in LLaMA for tasks contained in that paper, and taken from Wei et al. (2023) for other mathematical tasks.

**Datasets.** Following prior works, we evaluate on a number of datasets. The following tasks measure performance on algebraic word problems: **AQuA** (Ling et al., 2017), **ASDiv** (Miao et al., 2021), **GSM8K** (Cobbe et al., 2021), and **SVAMP** (Patel et al., 2021). We also evaluate on **MATH** (Hendrycks et al., 2021b), a larger and more challenging benchmark.

For commonsense reasoning, we measure open-ended performance on **CommonsenseQA** (Talmor et al., 2019) and **StrategyQA** (Geva et al., 2021). We also evaluate on a battery of multiple-choice reasoning benchmarks: both the easy and challenge splits of the **AI2 Reasoning Challenge** dataset (Clark et al., 2018), **BoolQ** (Clark et al., 2019), **HellaSwag** (Zellers et al., 2019), **MMLU** (Hendrycks et al., 2021a), **PIQA** (Bisk et al., 2019), **SIQA** (Sap et al., 2019), and **WinoGrande** (Sakaguchi et al., 2019).

## 3.2 Hyperparameter Selection

Contrastive decoding has three major hyperparameters: the masking ratio $\alpha$, the contrastive strength $\beta$ and the size of the amateur model. We find that results are fairly insensitive to $\alpha$ as long as $\beta$ is reasonably small (below 1); unless otherwise stated we use $\alpha = 0.1$ across experiments.

Next we consider the size of the amateur model. In agreement with Li et al. (2022), we find that performance benefits from smaller amateur models ( Figure 4); while a 1B-parameter amateur helps reasoning performance, a 7B-parameter amateur harms it. We also examine different types of amateurs; ablation studies show that a partially-trained amateur performs better than a fully-trained one, and that a poorly-prompted expert can be successfully used as an amateur as well (see subsection 4.2).

Finally, we examine the effect of $\beta$. The optimal value depends on the task, but for both generation tasks like GSM8K and multiple-choice ranking tasks like PIQA we find that $\beta = 0.5$ performs well. Setting $\beta$ too high can place too much weight in the contrastive penalty and harm performance, especially with a larger gap between amateur and expert models. $\beta = 0$ corresponds to standard greedy decoding with no contrastive penalty. Results of $\beta$ hyperparameter sweeps can be found in Table 1, Figure 4, Figure 5 and Appendix B.

The best result on GSM8K, with LLaMA-65B and $\beta = 0.25$, is 57.7 (Table 1), outperforming PaLM-540B (56.5), LLaMA-2 (56.8) and GPT-3.5 (57.1).* (Anil et al., 2023; OpenAI, 2023)

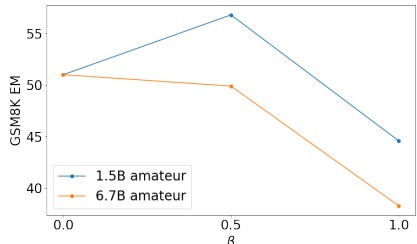

| Expert | $\beta = 0$ | $\beta = 0.25$ | $\beta = 0.5$ | $\beta = 1$ |
|--------|-------------|----------------|---------------|-------------|
| 7B | 10.7 | 11.5 | **13.6** | 11.0 |
| 13B | 17.0 | 21.0 | **22.9** | 20.4 |
| 30B | 35.2 | 40.0 | **43.4** | 42.0 |
| 65B | 51.0 | **57.7** | 56.8 | 44.6 |

Figure 4: Results on GSM8K with LLaMA-65B as the expert. While a 7B amateur harms performance, a 1.5B amateur helps.

Table 1: Results on GSM8K. $\beta = 0.5$ tends to give good results across expert sizes.

## 3.3 Arithmetic Reasoning

We find that contrastive decoding tends to help on arithmetic reasoning tasks with chain-of-thought prompting; see Table 2 for all results. One exception to this is the MATH dataset, which proves to be challenging for both standard and contrastive decoding. We conjecture that because contrastive decoding amplifies skills that the expert has learned better than the amateur, it cannot help on tasks that are well beyond the expert's ability.

We also experiment with normalizing the $\alpha$-masked CD scores via softmax, then temperature sampling from the resulting distribution. This permits CD to generate multiple candidate reasoning chains to be used for self-consistency (taking the majority answer) (Wang et al., 2023b). We show across both mathematical and commonsense reasoning, CD improves self-consistency performance.

---

*OpenAI (2023) evaluates GPT-3.5 5-shot; all others are 8-shot.

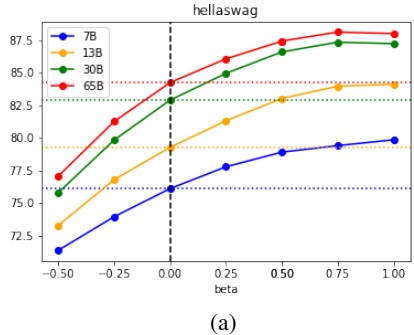
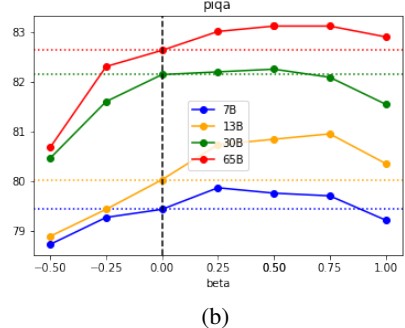

(a)                   (b)

Figure 5: Two examples of sweeping through $\beta$ values on multiple-choice reasoning tasks across model scales. Dashed horizontal lines mark performance without contrastive decoding.

Table 2: Results on math generation tasks. Contrastive decoding generally improves performance.

| Model | CD | AQuA | ASDiv | GSM8K | MATH | SVAMP | Average |
|---|---|---|---|---|---|---|---|
| 7B | ✗ | 21.0* | 40.2 | 10.7 | 3.0 | 27.3 | 20.4 |
| 13B | ✗ | 18.1* | 49.0 | 17.4 | 4.2 | 39.4 | 25.6 |
| 30B | ✗ | 23.8 | 60.1 | 35.3 | 6.9 | 55.9 | 36.4 |
| 65B | ✗ | 33.3 | 67.2 | 51.0 | 10.6 | 69.1 | 46.2 |
| 65B maj@20 | ✗ | 38.2 | 73.6 | 68.0 | –† | 77.3 | 64.3 |
| 7B | ✓ | 19.0* (-2.0) | 39.7 (-0.5) | 14.3 (+3.6) | 2.9 (-0.1) | 31.5 (+4.2) | 21.5 (+1.1) |
| 13B | ✓ | 16.0* (-2.1) | 52.0 (+3.0) | 22.7 (+5.5) | 3.8 (-0.4) | 43.1 (+3.7) | 27.5 (+1.9) |
| 30B | ✓ | 29.8 (+6.0) | 62.5 (+2.4) | 43.1 (+8.1) | 8.1 (+1.2) | 59.3 (+3.4) | 40.6 (+4.2) |
| 65B | ✓ | 36.9 (+3.6) | 71.9 (+4.7) | 56.8 (+5.8) | 10.3 (-0.3) | 67.8 (-1.3) | 48.7 (+2.5) |
| 65B maj@20 | ✓ | **39.4** (+1.2) | **77.4** (+3.8) | **74.0** (+6.0) | –† | **79.0** (+1.7) | **67.5** (+3.2) |

## 3.4 COMMONSENSE REASONING

Results are more mixed for CommonsenseQA and StrategyQA. For both of these tasks, we 8-shot prompt our model and compute the exact match score against the ground-truth answers. We find that contrastive decoding harms performance for smaller models, but that this harm equalizes somewhat for the 65B model and evens out when using self-consistency. See Table 3 for full results.

Table 3: CD harms commonsense reasoning with a smaller expert, but performance evens out with a larger expert-amateur gap.

| Model | CD | CSQA | StrategyQA | Average |
|---|---|---|---|---|
| 7B | ✗ | 40.0 | 59.2 | 49.6 |
| 13B | ✗ | 60.4 | 64.5 | 62.5 |
| 30B | ✗ | 66.4 | 68.7 | 67.6 |
| 65B | ✗ | 77.5 | 69.5 | 73.5 |
| 65B maj@20 | ✗ | 77.0 | **79.3** | 78.2 |
| 7B | ✓ | 37.3 (-2.7) | 58.3 (-0.9) | 47.8 (-1.8) |
| 13B | ✓ | 58.5 (-1.9) | 65.5 (+1.0) | 62.0 (-0.5) |
| 30B | ✓ | 62.8 (-3.6) | 67.6 (-1.1) | 65.2 (-2.4) |
| 65B | ✓ | 77.1 (-0.4) | 71.5 (+2.0) | 74.3 (+0.8) |
| 65B maj@20 | ✓ | **77.9** (+0.9) | **79.3** (+0.0) | **78.6** (+0.4) |

---

*In the AQuA task, the model selects one out of five given options. Thus the random baseline is 20%, and results below that threshold are not meaningful.

†Given the size of the dataset and length of generations, we do not evaluate maj @ 20 on MATH.

## 3.5 Contrastive Ranking

We further evaluate a contrastive objective as a scoring function to rank answers to multiple-choice questions. These tasks are zero-shot, multiple-choice cloze tasks; instead of open-ended generation the model scores each potential completion, length-normalizing following Touvron et al. (2023). We find comparable performance across most tasks, with more substantive gains on HellaSwag and ARC-Challenge. Notably, on HellaSwag CD leads LLaMA-65B to score 88.0, which outperforms LLaMA-2 (85.3), GPT-3.5 (85.5) (OpenAI, 2023) and PALM 2-Large (86.8) (Anil et al., 2023).

Table 4: Results on multiple-choice reasoning tasks. CD generally provides a modest boost.

| $\beta$ | ARC-E | ARC-C | BoolQ | HSwag | PIQA | SIQA | WGrande | MMLU | Avg |
|---|---|---|---|---|---|---|---|---|---|
| 0.0 | **79.1** | 56.1 | 84.2 | 84.2 | 82.6 | 52.3 | 77.3 | **63.5** | 72.4 |
| 0.5 | 79.0 | 59.5 | **84.3** | 87.4 | **83.1** | **53.3** | **77.8** | 63.4 | **74.9** |
| 1.0 | 76.9 | **59.7** | 84.1 | **88.0** | 82.9 | **53.3** | 76.5 | 63.2 | 74.5 |

## 4 Additional Studies

### 4.1 Effects of Contrastive Decoding

**CD is worse at arithmetic but better at logical reasoning.** We conduct a manual error analysis of 100 randomly selected examples from the GSM8K set between continuations from greedy decoding and CD ($\beta = 0.5, \alpha = 0.1$). We follow Wang et al. (2023a) and categorize wrong answers as primarily being due to an arithmetic error, a missing step or a semantic misunderstanding. We add one category of "degeneration," chosen when the model lapses into excessive repetition. Our small-scale analysis finds that CD makes more arithmetic errors, but that this is offset by better semantic reasoning and fewer missing steps (see Table 5).

Table 5: Proportion of errors in of a set of 100 GSM8K questions. CD makes more arithmetic errors, but omits fewer steps and avoids semantic misunderstandings.

| CD | Arithmetic | Missing Step | Semantic | Degeneration | Total Errors |
|---|---|---|---|---|---|
| ✗ | **4%** | 22% | 24% | 4% | 54% |
| ✓ | 8% | **20%** | **21%** | **3%** | **52%** |

To further explore the claim that the benefit of CD does not stem from arithmetic evaluation, we generate a toy dataset of 1,0000 multiplication and subtraction equations with operands up to four digits and then 8-shot prompt models to complete the expression, measuring exact match accuracy. We find that CD does not improve performance on this task, and in fact may degrade it slightly. Results are shown in Table 8.

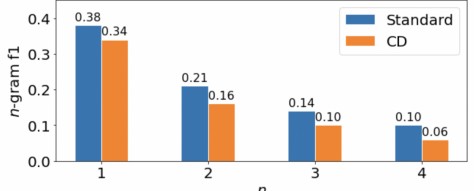

|  | Standard | CD |
|---|---|---|
| Correct % | 44.6 | **51.1** |
| Parseable % | 95.2 | **95.6** |
| Average # chars | 215.2 | 217.2 |

Table 6: High-level generation statistics from sampled generations on GSM8K. Responses are similar lengths, despite the performance improvement from CD.

Figure 6: CD reduces copying from the question in the generated Chain of Thought, as measured by n-gram overlap on GSM8K generations.

**CD reduces copying from the prompt.** We analyze 26,000 sampled generations from CD-sampling on GSM8K against the corresponding set from temperature sampling; both of these sets of generations are used in our self-consistency study. We find that responses are roughly the same length and follow the few-shot template roughly the same proportion of the time. This rules out the

hypothesis that contrastive decoding simply leads the model to follow the template better, prevents degeneration or induces longer answers with more reasoning steps. Further, we run an automatic evaluation of greedy generations using ROSCOE (Golovneva et al., 2022) but do not find significant differences in any of these metrics. However, we measure the precision and recall of the tokens in the prompt by the sampled generations and find that CD systematically reduces token-level copying from the prompt. This may be related to increased reasoning ability, as surface-level copying from the prompt does not provide new information to the problem.

**CD can harm factual recall.**    Our primary claim is that contrastive decoding improves chain-of-thought reasoning. However, we also test CD on two pure factual-recall tests that do not utilize chain-of-thought: OpenBookQA (Mihaylov et al., 2018) and TriviaQA (Joshi et al., 2017). Open-BookQA ("OBQA"), is a multiple-choice completion task, while TriviaQA is a 5-shot generation task. Reusing the same setup from reasoning leads to a slight degradation of performance, as seen in Table 7.

Table 7: CD can harm performance on factual recall tasks.

| CD | OBQA | TriviaQA* |
|----|------|-----------|
| ✗ | **60.0** | **72.2** |
| ✓ | 57.8 (-2.4) | 69.9 (-2.1) |

Table 8: CD slightly harms performance on a synthetic task of evaluating arithmetic expressions.

| CD | 7B | 13B | 30B | 65B |
|----|-----|------|------|------|
| ✗ | **31.0** | **36.3** | **52.3** | **58.4** |
| ✓ | 30.9 | 35.6 | 52.2 | 57.6 |

**CD outperforms other reasoning enhancements in FLOP efficiency.**    We note that contrastive decoding introduces relatively little overhead in comparison to other reasoning-enhancing methods. We estimate that with a 1.5B amateur and 65.2B expert, contrastive decoding increases the total number of FLOPs by $3.25\%$ (see section C of the appendix). This compares favorably to self-consistency, which requires several extra full generation loops. We show in Figure 9 that CD is significantly more efficient than self-consistency.

## 4.2 ABLATION STUDIES

$\alpha$**-masking alone is not enough.**    When sampling and performing self-consistency, $\alpha$-masking prevents the sampling of tokens the expert finds to be unlikely. It is natural to ask what portion of the benefit comes purely from $\alpha$-masking and not the contrastive objective itself.

To answer this, we set $\beta = 0$ but $\alpha = 0.1$; that is, we mask out candidates based on the expert but do not apply the contrastive objective. When sampling one path, we expect $\alpha$-masking to improve over temperature sampling alone as it eliminates unlikely results and thus provides a closer approximation to greedy sampling. This holds, but as we increase the number of paths we find no benefit from $\alpha$-masking alone. This suggests that the contrastive objective, and not $\alpha$-masking, is the primary source of improved self-consistency results. See Figure 7 for results of this ablation.

**CD requires chain-of-thought prompting to improve results.**    We next study whether contrastive decoding provides an advantage in the absence of chain-of-thought prompting. We remove the chains of thought from the GSM8K fewshot prompt, and find that as expected performance drops for both standard and contrastive decoding (Figure 8); further, without chains of thought contrastive decoding provides no consistent improvement. As with the MATH dataset, solving problems without explicit reasoning steps may be too challenging of a task for the expert model, and thus leave too small a gap between the expert and amateur to contrastively exploit.

**CD can benefit non-LLaMA models.**    We conduct a short study to show that CD can benefit models outside of the LLaMA family. For this study, we choose the FLAN-T5 family as it is open-source, has a wide range of model sizes that share a single tokenizer, and obtains good performance on chain-of-thought reasoning tasks. We use FLAN-T5-XXL (11B) as the expert model and FLAN-T5-Small (80M) as amateur. We evaluate on GSM8K using the 8-shot random prompts from Fu

---

*On manual examination, we find the set of correct answers provided by TriviaQA to be insufficient. Randomly sampling 100 supposedly incorrect answers generated by CD and standard decoding, we find roughly half are in fact correct (46/100 with CD and 49/100 without). A rough linear extrapolation gives us estimates for non-CD and CD scores of 85.8 and 83.7, respectively.

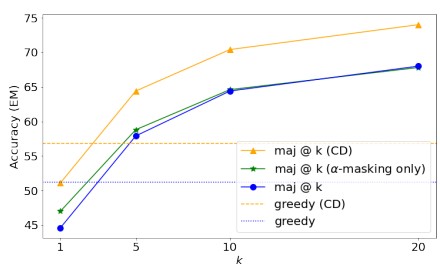

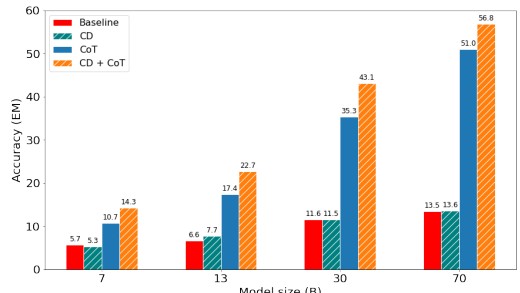

Figure 7: GSM8K scores via temperature sampling and maj @ $k$ with various values of $k$. $\alpha$-masking alone does not yield significant improvement, while full CD does.

Figure 8: Comparison of GSM8K scores with LLaMA-65B, both with and without chain-of-thought prompts. CD only helps when using CoT.

et al. (2023); note that GSM8K is within the set of tasks that FLAN-T5 is finetuned on. CD provides a slight boost in performance, as seen in Table 9. We leave more extensive experiments on other families of models to future work.

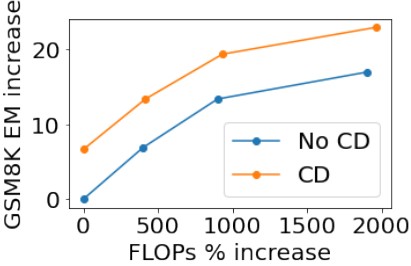

| CD | $\beta$ | GSM8K |
|----|---------|-------|
| ✗  | 0       | 16.4  |
| ✓  | 0.5     | 17.1  |
| ✓  | 1.0     | **17.4** |

Figure 9: FLOP increases, with increasing compute from using more samples for self-consistency. CD achieves similar or better performance with a smaller increase in FLOPs.

Table 9: FLAN-T5 performance on GSM8K. CD provides a boost to performance.

**Small-scale amateurs beat "negative prompting."** We experiment to determine if there is a more effective weak amateur model to use for contrastive decoding. We define a set of "negative prompts" by sampling 7B model outputs on the fewshot prompts and collecting the incorrect responses. We use these responses as fewshot prompts to mimic the failure modes of the family of models. These negative prompts should harm the performance of models they are prompted with, and specifically bias results towards the error distribution of the 65B model.

We find that contrasting with a negative prompt does not harm performance, but does not improve it as much as contrasting with a small amateur (see Table 10). In an ablation study, we find that negative prompting does not harm performance that much; prompting a 65B model with incorrect fewshot examples on GSM8K gives a score of 41.3, which underperforms prompting with correct examples (51.2) but significantly beats non-chain-of-thought prompting (13.5). This supports Wang et al. (2023a), who find that even incorrect chain-of-thought rationales improve reasoning. A prompting strategy which better incapacitates the expert model might yield better results.

**Mid-training checkpoints make for good amateurs.** We experiment with checkpoints of a mid-training 7B-parameter LLaMA model taken 10% and 23% of the way through the full training run. Even while a fully-trained 7B amateur harms performance on GSM8K, we find that a partially-trained amateur improves performance. We do not perform extensive hyperparameter sweeps here, instead reusing $\alpha = 0.1, \beta = 0.5$ as before. We do not pursue partially-trained amateurs for our main results as results may vary based on the order of training data, but this result allows us to interpret contrastive decoding as a first-order optimization step over the output of a model, highlighting the high-level behaviors that it learns later on in the course of training. See Table 11 for full results.

Table 10: On GSM8K, negative prompting outperforms greedy decoding but weakens CD.

| Expert | Greedy | NP | CD | CD + NP |
|--------|--------|------|--------|---------|
| 7B | 10.7 | 11.4 | **14.3** | 12.7 |
| 13B | 17.4 | 17.5 | **22.7** | 20.7 |
| 30B | 35.3 | 36.9 | **43.1** | 42.9 |
| 65B | 51.0 | 52.0 | **56.8** | 54.7 |

Table 11: Early-training checkpoints can be good amateurs, even when late-stage checkpoints harm performance.

| Amateur | Amateur Tokens | GSM8K |
|---------|---------------|-------|
| 7B | 130B | **57.0** |
| 7B | 300B | 56.8 |
| 7B | 1.3T | 49.9 |

## 5 RELATED WORK

**Steering methods for reasoning.** Other works more explicitly model the error distribution of reasoning steps and use this to steer decoding. For example GRACE (Khalifa et al., 2023) uses a contrastive loss to train an external step-level discriminator, which it then uses to select between candidate steps sampled from a base model. Using the interpretation of contrastive decoding as mutual distinguishability between amateur and expert, we see that our method is close to FUDGE (Yang & Klein, 2021) where the binary predictor is an estimate of the probability that the generated token has come from the expert rather than the amateur.

**Prompting Methods for Reasoning.** There are many recent prompting methods to improve language model reasoning; see Qiao et al. (2023) for a survey. We perform our experiments with chain-of-thought prompting (Wei et al., 2023).

**Sampling methods** Several decoding methods exist to improve the quality of generations from large language models. For open-ended generation, truncated sampling schemes like top-$k$ sampling (Fan et al., 2018), nucleus sampling (Holtzman et al., 2020) and typical sampling (Meister et al., 2023) have been shown to reduce repetition in comparison to greedy decoding and beam search while producing more coherent generations than standard temperature sampling. However, sampling can still introduce errors into logical chains, and so greedy decoding is used to more effectively solve reasoning tasks. (Wei et al., 2023; Anil et al., 2023)

**Contrastive Generation Methods.** Our formulation's objective can be interpreted as a special case of DExperts (Liu et al., 2021), using the larger model as both an expert and base LM prior. Yona et al. (2023) identify model biases with Contrastive Input Decoding, a contrastive-decoding-style technique similar to negative prompting that operates on perturbed text inputs.

Concurrently to our work, Chuang et al. (2023) propose DoLA, which improves factuality and reasoning through contrastive decoding between the predictions of later layers and earlier layers in a language model. We study a wider array of reasoning tasks and demonstrate that a 7B amateur is too large, finding greater gains in reasoning just by scaling down the amateur to 1.5B parameters.

Our paper differentiates itself from Li et al. (2022), which initially proposed Contrastive Decoding, in several ways: by testing on standard reasoning benchmarks, by our exploration of $\beta$ as a hyperparameter, by ablations with various types of amateurs, and by a careful analysis of the combination of Contrastive Decoding with chain-of-thought prompting and self-consistency.

## 6 LIMITATIONS

Our investigation is also limited mainly to the LLaMA family of models. While the method continues to provide benefit to larger LLaMA models, further work is required to definitively establish the effect of contrastive decoding on larger, tuned models.

## 7 CONCLUSION

Our study shows that contrastive decoding can improve chain-of-thought reasoning in large language models. While challenges like factual recall remain, this strengthens the case for contrastive decoding as a simple, general-purpose method to elicit more desirable behavior from large language models.

REPRODUCIBILITY STATEMENT

The training process and model architecture for the 1.5B-parameter LLaMA model used as the amateur in several results is publicly available, but the weights are not, which limits public reproducibility of results relying on that model. The results on FLAN-T5, as well as the negative-prompting study and examination of 7B-LLaMA as an amateur, are all built on entirely open-source models and data.

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

# A  APPENDIX

## A.1  CODE IMPLEMENTATION

We include PyTorch implementations of contrastive decoding in Algorithm 1 and Algorithm 2

---

**Algorithm 1:** Original formulation

```
# expert_logits - unnormalized scores from the expert model
# amateur_logits - unnormalized scores from the amateur model
# amateur_temp - temperature to normalize amateur distribution
# alpha - masking threshold

expert_probs = softmax(expert_logits, dim=-1)
amateur_probs = softmax(amateur_logits / amateur_temp, dim=-1)
cutoff = alpha*expert_probs.max(dim=-1, keepdim=True).values
diffs = log(expert_probs) - log(amateur_probs)
cd_logits = diffs.masked_fill(expert_probs < cutoff, -float('inf'))
```

---

**Algorithm 2:** Our formulation

```
# expert_logits - unnormalized scores from the expert model
# amateur_logits - unnormalized scores from the amateur model
# alpha - masking threshold
# beta - expert-amateur tradeoff parameter

cutoff = log(alpha) + expert_logits.max(dim=-1, keepdim=True).values
diffs = (1 + beta)*expert_logits - beta*amateur_logits
cd_logits = diffs.masked_fill(expert_logits < cutoff, -float('inf'))
```

## A.2 EQUIVALENCE - MASKING

For the sake of completeness, we show the equivalency between the two masking schemes. We restrict our consideration to the generation of a single token. Let

- $V$ be the vocabulary size of each model

- $\{e_i\}_{i=1}^V$ be the logits produced by the expert model

- $\{a_i\}_{i=1}^V$ be the logits produced by the amateur model

- $N_e$ and $N_a$ be the normalization terms of the softmax function; for example, $N_e = \sum_{j=1}^V \exp(e_i)$

The probability that the expert assigns to token $i$ after the softmax is by definition $p_e(i) = \frac{\exp(s_i)}{N_e}$

Now consider any token that is masked out. We have that $p_e(i) < \alpha * p_e(i_{max})$, where $i_{max}$ is the the token that maximizes $p_e$.

Because $e^x$ and $\log x$ are both strictly increasing functions, we obtain:

$$s_i - \log N_e < \log \alpha + s_{max} - \log N_e$$
$$s_i < \log \alpha + s_{max}$$

These two conditions are equivalent, and so we can mask tokens by thresholding their logits against $\log \alpha + s_{max}$.

Further, let us introduce an expert temperature parameter $\tau \in (0, \infty)$ that scales the logits arbitrarily. Then we obtain a new set of logits $c_i = \frac{s_i}{\tau}$

By then substituting $\alpha_\tau = \alpha^{1/\tau}$, we obtain the same mask. Thus the mask is a function that depends only on the quantity $\tau \log \alpha$, or equivalently $\alpha \exp(\tau)$. As we later show, we can fix $\tau = 1$ by introducing a new hyperparameter $\beta$. So if we fix $\tau = 1$, then our mask depends solely on $\alpha$.

Letting $p_e^{(i)}$ correspond to the post-softmax probability that the expert assigns to token $i$, we have shown that the valid set produced by our method is the same as in the original:

$$V_{valid} = \left\{ j \in V, p_e^{(j)} \geq \frac{1}{\alpha} \max_{k \in V} p_e^{(k)} \right\}$$

## A.3 EQUIVALENCE - LOGIT COMBINATION

To be concise, first define $q(i) = \frac{p_e(i)}{p_a(i)}$ be the ratio of the probability assigned by the expert model over the probability from the amateur model, both on token $i$.

Further, let $s_i$ denote the value of the logit that CD assigns to token $i$. Then

$$s_i = (1 + \beta) \log p_e(i) - \beta \log p_a(i)$$
$$\exp(s_i) = p_e(i) q(i)^\beta$$
$$p_{cd}(i) \propto p_e(i) q(i)^\beta$$

Expanding this, we obtain:

$$p_{cd}(i) \propto \exp\left((1 + \beta)e_i - \beta a_i\right)$$

which is equivalent to simply linearly combining the expert and amateur logits.

When we introduce temperatures, the equation becomes

$$p_{cd}(i) \propto \exp\left(\frac{1+\beta}{\tau_e}e_i - \frac{\beta}{\tau_a}a_i\right)$$

$$\text{mask}(i) = f(\tau_e \log \alpha)$$

When sampling, the temperature with which we sample from the CD logits is introduced as $\tau_{out}$

$$p_{cd}(i) \propto \exp\left(\frac{1}{\tau_{out}}\left(\frac{1+\beta}{\tau_e}e_i - \frac{\beta}{\tau_a}a_i\right)\right)$$

These four parameters $\tau_{out}$, $\tau_e$, $\tau_a$ and $\beta$ combine into only two coefficients – one for $e_i$ and one for $e_i$.

$$p_{cd}(i) \propto \exp\left(\kappa_e e_i - \kappa_a a_i\right)$$

We now fix $\tau_a$ and $\tau_e$ to 1. We can obtain almost the same range of values with the $\tau_{out}$, $\beta$ formulation as with the $\kappa_e$, $\kappa_a$ formulation. The only exception is the case for which $\kappa_e = \kappa_a$, which we exclude after finding that weighing expert and amateur equally gives worse results than down-weighing the amateur. Despite this exception, we prefer the $\beta$, $\tau_{out}$ formulation because it decouples the scale of the final logits from the $\beta$ parameter: in expectation $\beta$ does not scale the logits up or down, and so when sampling $\tau$ will affect generation diversity and $\beta$ will affect the expert-amateur tradeoff.

# B MULTIPLE-CHOICE BETA SWEEP RESULTS

Here we include the plots for all beta sweeps through the multiple-choice tasks.

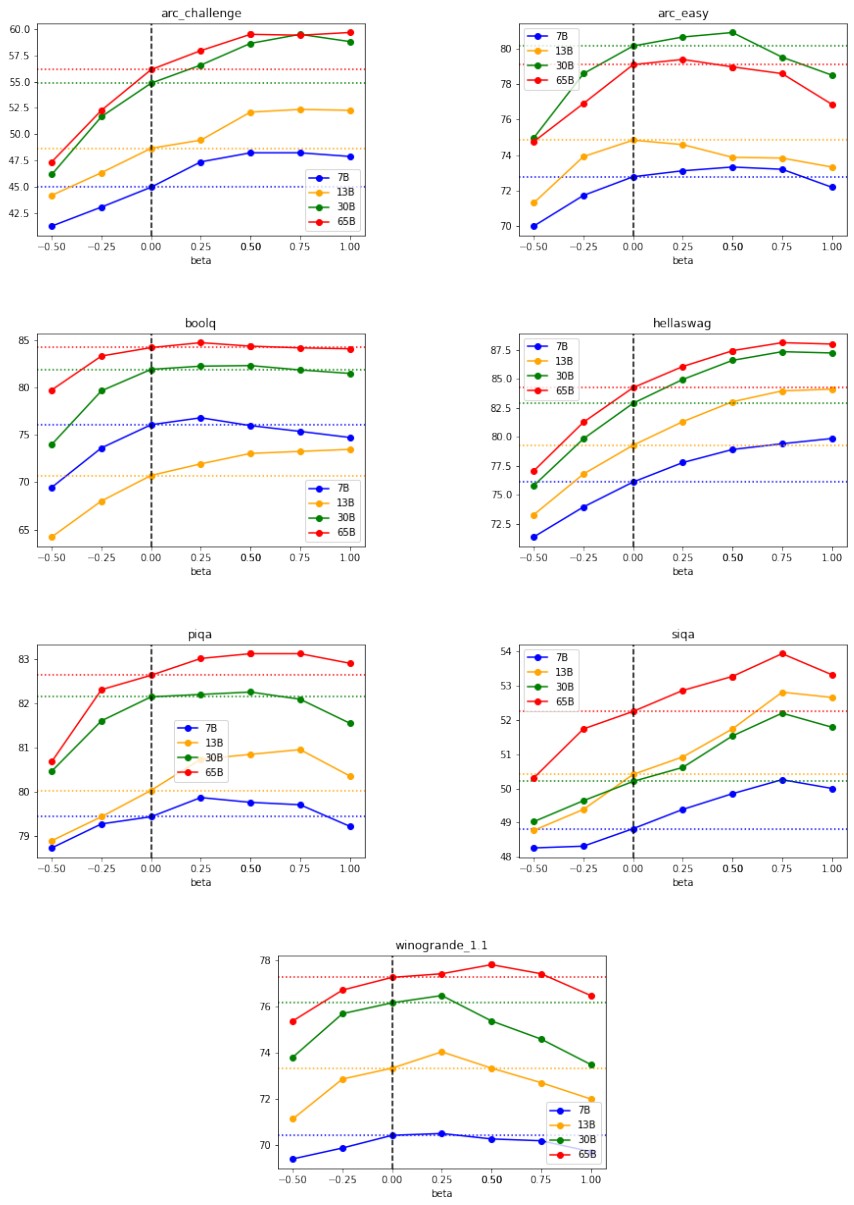

## C  FLOP ESTIMATES

We follow Kaplan et al. (2020) and estimate the number of flops for one forward pass of a Transformer model with $N$ non-embedding parameters as roughly $2N$. For the purposes of this analysis, we drop the negligibly small $2n_{layer}n_{ctx}d_{attn}$ term. This allows us to consider the cost of generating one token to be constant regardless of the context length.

With this approximation, we find that contrastive decoding adds $\frac{1.5}{65.2} \approx 2.30\%$ to the number of FLOPs per forward pass for a 65B expert model. To ensure a fair comparison, we also include the small increase in generation lengths induced by CD (see Table 6), bringing its total percent increase up to $3.25\%$.

