# OpenReview forum: "Contrastive Decoding Improves Reasoning in Large Language Models"
_ICLR.cc/2024/Conference — Submitted to ICLR 2024_

### Official Review · Reviewer_ee57 · 2023-10-27

**Soundness:** 3 good
**Presentation:** 3 good
**Contribution:** 3 good
**Rating:** 5
**Confidence:** 5

**Summary:**

This paper explores how well Contrastive Decoding (CD), an alternative generation strategy proposed by Li et al. in 2022, performs on reasoning tasks. It reformulates CD and introduces a hyperparameter \beta, which controls the strength of the contrastive penalty. Contrastive Decoding (CD) can significantly improve over greedy decoding on some reasoning tasks, especially arithmetic tasks such as GSM8K. It can also yield smaller gains on commonsense reasoning tasks with large models and some contrastive ranking tasks such as HSwag. However, CD can degrade performance on other tasks, such as MATH and commonsense reasoning tasks with smaller models. Relative to greedy decoding, CD performs worse on arithmetic skills, better on logical reasoning, and can harm factual recall.
The paper shows that CD works better under certain conditions, such as when using weaker amateur models and with chain-of-thought prompting. Although the paper reports most of its experiments on LLAMA models, it does show that CD can produce small gains on FLAN-T5 in one experiment.

**Strengths:**

* Shows that Contrastive Decoding can improve upon greedy decoding for some reasoning tasks
* Reports ablation studies exploring the type of problems where CD performs better or worse than greedy decoding.
* Presents ablations that show the factors that affect the performance of CD, such as the sensitivity to the contrastive penalty term (i.e., β) and the type of amateur model (e.g., weak vs. strong, using negative prompting, using earlier checkpoints).

**Weaknesses:**

* Contrastive Decoding (CD) yields highly inconsistent results. For example, it improves on arithmetic reasoning tasks such as GSM8K but not on MATH. It yields small gains on Commonsense Reasoning tasks, but only with large models and on some Contrastive Ranking tasks such as HSwag. Given these mixed results, it is unclear when to use CD effectively.
* The paper does not provide a detailed error analysis by presenting wins/losses on tasks or offer other insights into why CD performs better/worse on specific tasks.
* Contrastive Decoding (CD) only works with chain-of-thought reasoning. This seems like a significant limitation, and it would be valuable to conduct more ablation studies to understand it better.
* Most of the experiments in the paper (except for the single T5-FLan result) are obtained with LLAMA models. This makes it unclear whether CD would show the same behavior with other pre-trained models.

**Questions:**

* It would be good to provide error analysis (samples of wins/losses) especially when CD does not work e.g. on MATH
* In 2.1, it would be good to highlight that \alpha in [0,1] i.e. log \alpha <0.
* Figure 4: What does 'GSM 8K EM' mean?
* Table 1: What is the metric being shown?
* P7: In the section titled 'CD reduces copying from the prompt', can you provide statistics from the experiment for the baseline/CD models?
* Sec 2.2: It is not immediately obvious why the distribution in Equation 1 collapses to the argmax of pe/pa as beta->\infinity. It would be useful to show the steps in the appendix.

---

> ### Author Response · Authors · 2023-11-21
>
> Thank you for your detailed review and feedback! We have updated the paper to address some of your concerns, and are open to further discussion and changes as need be.
>
> ### Responses to Weaknesses:
> - **Consistency of results:** We have highlighted that contrastive decoding yields consistent improvements when coupled with self-consistency on all measured tasks, providing at least one generation paradigm in which it is always shown to help. Further, we include analysis for the generation tasks on which contrastive decoding fails to improve performance -- for instance, the smaller expert models perform worse than chance on AQuA and therefore do not have a clear advantage over the amateur model to contrastively exploit, and the expert’s overall performance on the MATH benchmark is quite low. We also now include more analysis in Section 3.5 about the challenges of CD on ARC-Easy (inconsistent scaling behavior) and MMLU (factual-heavy).
> - **Error analysis:** We now include example generations with and without CD in the appendix of the paper. Most differences are difficult to attribute to CD for reasonable values of beta, and so we rely on the differences in automatic metrics (e.g. n-gram copying) as well as the manual error analysis in Table 5 to give insight into contrastive decoding’s strengths and weaknesses.
> - **Chain-of-Thought Limitation:** We now clarify in both Section 3.3 and 4.2 that contrastive decoding only works when the expert is significantly better than the amateur at a given task. Previous work from Li et al find benefits to open-ended text generation, and our experiments show that non-chain-of-thought performance does not improve as much as chain-of-thought performance with scale, suggesting a lack of a sufficient contrastive gap to exploit. In other words, solving these tasks effectively with an untuned model requires chain-of-thought generations both with and without contrastive decoding.
> - **LLaMA Models:** Contrastive decoding, and contrastive-decoding-like strategies like coherence boosting (Malkin et al 2022) and DoLa (Chuang et al 2023) have been shown to improve performance on open-ended task generation. Most of these works use one to two families of models; we include the FLAN-T5 study to show that our method works out-of-the-box on a family of models without tuning any additional hyperparameters. Further, the original Contrastive Decoding work found improved open-ended generation on OPT and GPT-2, suggesting that the efficacy of scale-based Contrastive Decoding is not unique to the LLaMA family of models.
>
> ### Responses to Questions
> - We have now included example generations in the appendix for qualitative analysis. It is important to note that CD does not significantly harm performance on the MATH benchmark (a maximum of -0.4% absolute performance), even though it does not bring the same benefit it does to other arithmetic reasoning tasks.
> - We have refactored 2.1 to give more background on the original method, and in doing so emphasize the range of possible values for alpha.
> - In this context, GSM8K “EM” is the exact match accuracy on GSM8K results. For clarity, we have changed the label to “Accuracy (GSM8K)”.
> - Similarly, we have changed the caption to read “Accuracy on the GSM8K arithmetic reasoning benchmark.”
> - We include the n-gram F1 scores for both greedy decoding (blue bars) and contrastive decoding (orange bars), with greedy decoding as the baseline in Figure 6.
> - As $\beta$ goes to $\infty$, the ratio term dominates the the prior term, assuming there is at least one item in the feasible set which the expert assigns higher probability than the amateur (creating a ratio greater than one which the beta exponent would amplify infinitely above the rest).

---

> ### Comment · Reviewer_ee57 · 2023-11-22
> **Thanks for your clarifications**
>
> Thank you for providing the clarifications.

---

### Official Review · Reviewer_CBdx · 2023-10-31

**Soundness:** 2 fair
**Presentation:** 3 good
**Contribution:** 2 fair
**Rating:** 5
**Confidence:** 4

**Summary:**

The paper addresses reasoning ability of large language models. The authors exploit and slightly modify and existing approach (Contrastive decoding) to improve, at inference time, the reasoning ability of pretrained LLMs.  The experimental results give a thorough analysis of the power of this approach, for long-form generation and for reasoning tasks.

**Strengths:**

1) The paper reports exhaustive experimental results (arithmetic analysis, commonsense reasoning, ranking), parameters analysis and ablation studies. Quantitative analysis and comparisons with related methods are well presented. I appreciate the overall discussions of the strengths and weaknesses of the model in different scenarios/case studies (ie, arithmetic vs reasoning). The overall conclusions of this experimental results highlight the potential of the Contrastive decoding approach and might be exploited in future work by others.

2) The paper is well written overall, the general purpose of the approach clearly presented.

**Weaknesses:**

1) Novelty
The experimental results and conclusions of the paper are certainly interesting, but those are mainly based on an idea and concept borrowed from an existing paper (Li and al, 2022). The changes wrt to this previous work are minor in my opinion. The strength of the current submission is the experimental nature of the work, but the approach itself lacks of novelty.

2) Presentation (minor weakness)
Since the current approach is based on previous work (Contrastive decoding, Li and al, 2022), it would be preferable to first give a recall of this early work, before introducing the revised formulation proposed by the authors. I had to read the original work in order to understand section 2.1, the meaning of the alpha-mask and of logic/motivation behind the logits-substraction.

**Questions:**

Could we use the idea of Contrastive decoding during the training stage itself?

**Details Of Ethics Concerns:**

The current work does not raise ethical red flags wrt  state-of-the-art LLMs.

---

> ### Author Response · Authors · 2023-11-21
>
> Thank you for your detailed review and feedback! We have updated the paper to address some of your concerns, and are open to further discussion and changes as need be.
>
> ### Response to Weaknesses
> **Novelty**
>
> While the central method of our paper draws on Li et al 2022, we believe that our reformulation of the contrastive objective and application to reasoning tasks constitutes a novel contribution that fills a clear research gap.
>
> Contrastive decoding was originally proposed as an alternative to sampling methods in order to improve open-ended text generation in language models. Given a single generation, most decoding methods that produce more diverse text (e.g. temperature sampling, nucleus sampling) harm performance on reasoning tasks. In showing that contrastive decoding improves reasoning on performance tasks, our findings shift a common understanding of decoding methods: that there is a hard tradeoff between diversity and reasoning ability.
>
> Beyond our findings, our reformulation constitutes a novel benefit to research on contrastive decoding in the future. The application of too-large amateurs under the original, non-$\beta$ formulation of CD has led other papers to wrongly conclude that contrastive decoding harms reasoning. (DoLa; Chuang et al 2023) The $\beta$-parameter formulation allows better fine-grained control over CD, allowing a continuous reduction to standard greedy decoding in a way that the original formulation does not and making it more straightforward to apply to most problems (with a reasonable search space of [0, 1] rather than [1, $\infty$]).
>
> Besides this reformulation, we combine self-consistency with contrastive decoding for the first time, finding greater benefits than either alone. Negative prompting and contrast against a partially trained amateur are both novel methods that improve results as well.
>
> Thus, we believe that our paper presents several novel methods to study, in addition to a key reformulation of a previous work as well as counterintuitive findings that fill a clear research gap.
>
> **Presentation:** Thank you for this feedback -- for clarity, we have expanded section 2.1 with more background on the original method prior to our reformulation.
> ### Response to Questions
> - Yes; distilling the contrastive objective into a single language model is an ongoing work. It remains out of scope for the purposes of this paper, which focuses on merits of the training-free, inference-time intervention form of contrastive decoding as it pertains to reasoning tasks.
>
> Again, thank you for your feedback to help us make the paper clearer. Please let us know if this helps to address your concerns, or if there remain sections of the paper that remain unclear, and we would be happy to update the paper accordingly.

---

> > ### Comment · Reviewer_CBdx · 2023-11-23
> >
> > Thank you for the clarifications and for the improvement of section 2.1. Despite very interesting experimental conclusion, I do believe that the authors need to better expose the novelty of the work.

---

### Official Review · Reviewer_Y5Km · 2023-11-01

**Soundness:** 3 good
**Presentation:** 3 good
**Contribution:** 1 poor
**Rating:** 3
**Confidence:** 4

**Summary:**

The authors experiment with Contrastive decoding, proposed by Li et al 2022, on three groups of reasoning tasks: algebraic word problems, commensense reasoning, and multiple choice reasoning. Contrastive decoding improves generations by finding sequences which maximizes a likelihood difference between a “strong” and “weak” model. They report better performances compared to greedy decoding for most of the arithmetic tasks and mixed results for commensense reasoning tasks. The authors analyze the out-of-the-box gain in performance and state that it could be attributed to contrastive decoding’s ability to prevent unwanted patterns in the strong model’s output (e.g. generic phrases, copies of the input phrases, etc) compared to greedy generation.

**Strengths:**

The refactoring of the original contrastive decoding formulation to work in logit space is a nice idea. Authors claim that this makes the method more interpretable, which I’m not sure about.

The authors cover a good number of tasks for arithmetic, commensense and multiple-choice reasoning. This makes for interesting results in the additional studies section, in which they did a good job at analyzing generations from CD to interpret the gains or losses from CD compared to greedy generation.

**Weaknesses:**

The paper lacks novelty as its primary contribution is merely the application of an existing method to additional datasets without introducing innovative or original elements.

The paper’s concept bears a noticeable resemblance to $\href{https://aclanthology.org/2022.acl-long.565}{Coherence Boosting}$ by Malkin et al 2022, which is similarly a simple inference-time method improving generations and rankings. However, this paper is not mentioned in related works. It seems that it would also be a reasonable baseline to compare contrastive decoding to.

The mixed results on commensense reasoning tasks, and results without self-consistency on arithmetic datasets do not strongly support the claims, and do not align well with the narrative and title of the paper.

**Questions:**

As mentioned before, the method is very similar to coherence boosting (Malkin et al 2022). I believe it is reasonable to add it as a baseline and compare results.

In all the experiments (except for FLAN-T5), the smallest amateur model has 1B parameters. What would happen if you used smaller models? Given that CD can benefit from mid-training checkpoints, I believe studying smaller models as amateur models would be interesting. Perhaps you could use a different family of models and experiment with cross-family “strong” and “weak” models.

If contrastive decoding improves overall generation quality, it should ideally exhibit some improvement in the results without the presence of CoT in the prompts. Do you have any insights on why this is not happening?

In Fig.9 the FLOP increase of self-consistency is compared to contrastive decoding. Self-consistency does not have a hyper-parameter. How would you factor in your hyper-parameters (and search) in the FLOPS increase for CD?

Not a question, but I think that Fig.1 could benefit from an “Average” row at the top as well (at least the greedy generations).

---

> ### Author Response · Authors · 2023-11-21
>
> Thank you for your detailed review and feedback! We have updated the paper to address some of your concerns, and are open to further discussion and changes as need be.
>
> ### Response to Novelty
>
> While the central method of our paper draws on Li et al 2022, we believe that our reformulation of the contrastive objective and application to reasoning tasks constitutes a novel contribution that fills a clear research gap. Given a single generation, most decoding methods that produce more diverse text (e.g. sampling) harm reasoning. In showing that CD improves reasoning, our findings contradict this supposed tradeoff.
>
> Beyond our findings, our reformulation provides a novel benefit. The application of too-large amateurs under the original, non-$\beta$ formulation of CD has led other papers to wrongly conclude that CD harms reasoning. (DoLa; Chuang et al 2023) The $\beta$-parameter formulation allows more fine-grained control over CD, allowing a continuous reduction to standard greedy decoding in a way that the original formulation does not and making it more straightforward to apply to most problems (with a reasonable search space of [0, 1] rather than [0.5, $\infty$]).
>
> Besides this reformulation, we combine self-consistency with CD for the first time, finding greater benefits than either alone. Negative prompting and contrast against a partially trained amateur are also both novel methods that we find to improve results.
>
> Thus, we believe that our paper presents several novel approaches to study, a key reformulation of a previous work and surprising findings that fill a clear research gap.
>
> ### Responses to Other Strengths and Weaknesses
>
> - **Interpretability:** Our formulation replaces two opaque hyperparameters (amateur & expert temperatures) with one that cleanly corresponds to contrastive strength, which is orthogonal to the choice of output temperature. It also allows a continuous reduction to greedy decoding (by sending $\beta$ to zero).
> - **Coherence boosting:** Thank you for drawing our attention to this paper. Our paper now references coherence boosting and several other CD-inspired papers (CID, CAD). CB benchmarks on some shared ranking tasks as our study in Table 4, although we do not optimize beta on every task in the same way so as to avoid overfitting to each benchmark. Our central results, on generation tasks for reasoning, remain unique.
> - **Inconsistent results:** It is difficult to extract a trend from StrategyQA and CSQA, both of which are noisy and at times hurt or helped by CD. CSQA in particular behaves oddly in some ways; for example, maj @ 20 only boosts performance 0.9%. CD is also weak at factual recall, which is somewhat entangled with commonsense reasoning and could help explain the gap between performance on mathematical and commonsense tasks. Still, the aggregate trend with CD is relatively neutral for commonsense and positive for mathematical reasoning.
>
> ### Responses to Questions
> - **Coherence boosting:** This is a good suggestion for a baseline, but the hyperparameter selection of $\alpha$ and the context window $k$ is too costly to optimize and then benchmark effectively during the rebuttal period; note that coherence boosting is much more costly in FLOPs than CD as well, and so it isn't a completely fair comparison from an efficiency standpoint.
> - **Smaller Models:** We are also interested in studying smaller models. Cross-family models are possible, although it requires aligning different tokenizer vocabularies.
> - **Non-CoT Results:** There is not as large of an expert-amateur gap without CoT; further, one-step QA is a poor benchmark for a decoding method. CD’s main benefits (fewer skipped steps, fewer semantic errors) are not helpful without explicit reasoning steps. We notice a similar result for factual recall, which CD harms.
> - **FLoPs:** **Importantly, we use a single $\beta$ value (0.5) across all tasks, after optimizing it only on GSM8K.** Thus our results do not rely on additional compute to search for an task-optimal $\beta$. This deflates our results; we could likely achieve better per-task performance by more carefully fitting $\beta$. However, our results show that a user of CD could find benefits across many tasks with one choice of $(\alpha, \beta)$.
> - **Figure 1:** Thank you for the suggestion. We have added per-task averages to Figure 1.
>
> Again, thank you for your feedback -- we hope that this addresses some concerns you had with the paper. Please let us know if there is anything that remains unclear and we would be happy to update the paper accordingly.

---

> > ### Comment · Reviewer_Y5Km · 2023-11-23
> > **Thanks for your reply**
> >
> > Thank you for replying and providing clarifications.
> > While the reformulation is a nice idea I still believe the that the paper lacks novelty. It is good to hear that you agree on extending the work to smaller models.

---

### Meta-Review · Area_Chair_Km3A · 2023-12-07

**Metareview:**

This paper investigates the application of Contrastive Decoding, a simple text generation method, to improve performance on various reasoning tasks, demonstrating improvements on several benchmarks. After careful reading of the reviews, I would not recommend to accept the paper of its current form due to several concerns, including a lack of novelty as it primarily applies an existing method without introducing innovative elements, resemblance to another work without appropriate acknowledgment or comparison, and mixed results on commonsense reasoning and arithmetic tasks that do not strongly support the claims. Furthermore, the method's effectiveness with other pre-trained models remains unclear and can be improved from a larger scope of experiments.

**Justification For Why Not Higher Score:**

n/a

**Justification For Why Not Lower Score:**

n/a

---

### Decision · Program_Chairs · 2024-01-16

Reject